# Learning Generalized Trackers with Elastic Token Budgets

Yinchao Ma [* 1]  Jianpeng Yang [* 1]  Yuyang Tang [1]  Jie Xiao [1]  Dengqing Yang [1]  Tianzhu Zhang [† 1]

## Abstract

Visual tracking aims to estimate target states in video sequences, with applications spanning diverse computational requirements. Recent methods optimize trackers using manually pruned image tokens with a fixed budget to reduce computational costs. However, these trackers, once trained, are constrained to perform tracking under a fixed computational budget, limiting their adaptability to real-world computational diversity. To address the above limitation, we provide the first exploration of the elastic token budget training framework (ETBTrack), enabling trackers to perform robust tracking under varying computational budgets. It enjoys several merits. First, we present a novel result-driven importance criteria, in which we optimize a policy network guided by the localization precision of the tracker to estimate token importance, thereby aligning the objectives of importance estimation and tracking precision. Second, we develop a new budget-collaborative optimization strategy, in which we collaboratively optimize the tracker across varying budgets, thereby enabling the tracker to be compatible with diverse budgets. Two optimization processes are performed alternately to enhance the capability of elastic inference. Extensive experiments on large-scale benchmarks demonstrate the effectiveness of our method.

## 1. Introduction

Visual tracking constitutes a fundamental research area in computer vision, focused on automatically localizing the given reference object in a video sequence (Yilmaz et al., 2006). It has been successfully deployed in practical applications spanning diverse computational requirements, such as

*Equal contribution †Corresponding author ¹School of Information Science and Technology / National Key Laboratory of Deep Space Exploration, University of Science and Technology of China. Correspondence to: Tianzhu Zhang <tzzhang@ustc.edu.cn>.

*Proceedings of the 43rd International Conference on Machine Learning*, Seoul, South Korea. PMLR 306, 2026. Copyright 2026 by the author(s).

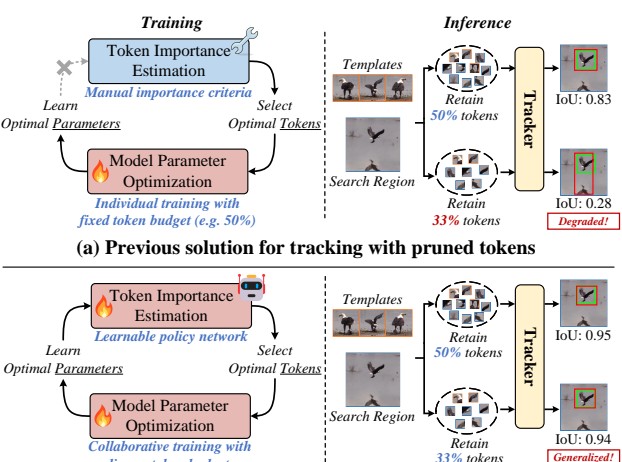

*Figure 1.* (a) Recent methods optimize the parameters of trackers using manually pruned image tokens with a fixed budget to reduce computational costs, which are constrained to perform robust tracking under the specific budget. (b) ETBTrack alternately optimizes the selected tokens and model parameters for diverse token budgets to find the optimal solution for generalized elastic tracking.

drones, robotic systems, and autonomous platforms (Zhang et al., 2013; Liu et al., 2015; Ma et al., 2024a; Tang et al., 2025b; Ma et al., 2025; Xiao et al., 2025; Tang et al., 2025a). Despite significant advancements in recent years, developing a tracker capable of robust tracking under diverse computational demands still faces substantial challenges.

Thanks to their global modeling capability and architectural flexibility, Transformer-based trackers (Ye et al., 2022; Yutao et al., 2022) have become the dominant paradigm in visual tracking research. Typically, Mixformer (Yutao et al., 2022) introduces the Transformer-based framework for joint feature extraction and interaction. ODTrack (Zheng et al., 2024) incorporates temporal information through concatenating online templates and temporal tokens into Transformer inputs. Further, due to the computational costs of attention mechanisms in Transformers scaling quadratically with the number of input tokens, some works (Ye et al., 2022; Xu et al., 2025; Zhou et al., 2023) make efforts to prune unimportant image tokens using manual criteria. OSTrack (Ye et al., 2022) progressively prunes search region tokens based on attention scores, and LMTrack (Xu et al., 2025) prunes reference tokens in memory according to the

aggregation of classification scores and attention scores. Despite these advances, they typically prune tokens to a fixed budget for both training and inference, as shown in Figure 1(a), which pose challenges in flexible computing and deployment scenarios.

To address the above limitations, we seek to develop a generalizable post-training framework for trackers that enables robust tracking under elastic token budgets to satisfy diverse computational demands. By studying previous methods, we summarize two critical issues that need to be considered. **1) How to identify the most critical tokens for tracking**. The key challenge in achieving robust tracking with elastic token budgets stems from the accurate estimation of token importance for precise target localization. Previous algorithms (Ye et al., 2022; Xu et al., 2025) employ manual importance criteria to prune tokens, which inevitably introduces potential biases and leads to suboptimal results during elastic inference. To address this limitation, we seek to directly optimize a lightweight policy network guided by the localization precision of the tracker to identify the most critical tokens, thereby mitigates potential manual biases. **2) How to optimize the tracker for elastic token budgets**. The parameters of existing trackers (Zhou et al., 2023; Ye et al., 2022; Zheng et al., 2024) are optimized under a fixed token budget (full tokens or pruned tokens), whose performance remains limited when evaluated with a different token budget than that used during training, even if the optimal tokens are retained. Therefore, further parameter optimization is required to improve the performance upper bound of the tracker under varying token budgets. However, directly employing alternating training with varying token budgets is prone to training instability and capability bias. Therefore, it is essential to explore a more stable optimization strategy for training with elastic token budgets.

Based on the above discussion, we propose a novel post-training framework to enable trackers with elastic token budgets, termed ETBTrack. ETBTrack is universally compatible with Transformer-based tracking architectures. Specifically, ETBTrack comprises two alternating optimization processes, as shown in Figure 1(b). **1) Token importance estimation.** Unlike previous manual importance criteria, we present a novel result-driven importance criteria. Specifically, we directly optimize a lightweight policy network guided by the localization precision of the tracker to rank and select the most critical tokens, thus mitigating the potential bias that could be introduced through manual importance criteria. **2) Elastic inference enhancement**. We develop a new budget-collaborative optimization strategy, in which the tracker is optimized by collaborative training with different token budgets to identify a common optimization direction. This enables the tracker to be compatible with diverse computational budgets. Notably, SeqTrack-L256 (Xin et al., 2023) trained using ETBTrack achieves 99.5% aver-

age performance with 33.3% tokens, 37.7% peak memory, and 2.0× inference speed.

To summarize, the main contributions of this work are: (1) To our knowledge, we provide the first exploration of the elastic token budget training framework (ETBTrack), enabling trackers to perform robust tracking under varying computational budgets. (2) We present a novel result-driven importance criteria, which estimates the importance of tokens guided by the tracker localization precision. We develop a new budget-collaborative optimization strategy, enabling the tracker to be compatible with diverse computational budgets. (3) Extensive experimental results demonstrate that ETBTrack enables trackers to perform robust tracking under varying token budgets.

## 2. Related Work

In this section, we overview visual tracking methods from network architectures and acceleration algorithms.

### 2.1. Network Architecture for Visual Tracking

Popular visual trackers can be broadly categorized into CNN-based trackers and Transformer-based trackers.

**CNN-based Tracker**. SiamFC (Bertinetto et al., 2016) pioneers the use of CNN-based siamese networks for feature matching and localization. SiamRPN++ (Li et al., 2019) extends the feature matching operations to depth-wise cross-correlation and localized targets through regression and classification. Moreover, DiMP (Bhat et al., 2019) trains a convolution filter using online tracking samples to distinguish the target object from background regions for localization. Despite significant progress (Voigtlaender et al., 2020; Mayer et al., 2021; Cui et al., 2022), the performance of CNN-based trackers remains constrained due to limited receptive fields and linear interaction mechanisms.

**Transformer-based Tracker**. With the advancement of Transformers (Vaswani et al., 2017), numerous trackers (Chen et al., 2022; Ma et al., 2024b; Wei et al., 2023) have been proposed that leverage the global non-linear modeling capabilities and architectural flexibility to improve tracking performance. Typically, MixFormer (Yutao et al., 2022) and OSTrack (Ye et al., 2022) propose Transformer-based joint feature extraction and interaction frameworks for visual tracking, achieving superior performance. SeqTrack (Xin et al., 2023) and ARTrack (Wei et al., 2023) design autoregressive tracking frameworks based on causal Transformer to enable precise bounding box regression. ODTrack (Zheng et al., 2024) and MCITrack (Kang et al., 2025) introduce temporal propagation modules and incorporate multiple online templates to capture temporal dynamics. Despite these advances, these trackers typically incur high computational overhead and lack the flexibility

to meet the diverse computational requirements in practical applications.

## 2.2. Acceleration Algorithm for Visual Tracking

Numerous works make efforts to accelerate the tracking process through network pruning or token pruning.

**Network Pruning**. Many works (Zhang et al., 2025; Yang et al., 2025b) accelerate tracking by pruning network structures. Typically, LightTrack (Yan et al., 2021) employs neural architecture search to identify promising architectures for efficient tracking. MixFormerV2 (Yutao et al., 2024) utilizes model distillation to gradually eliminate Transformer layers, thereby reducing inference latency. SGLATrack (Xue et al., 2025) dynamically disables representation-similar layers to achieve a precision-speed trade-off. However, once trained, these trackers cannot scale dynamically to meet diverse computational and precision requirements.

**Token Pruning**. With the popularity of the Transformer architecture, token pruning has attracted increasing research attention. Typically, OSTrack (Ye et al., 2022) and ODTrack (Zheng et al., 2024) reduces computational cost by progressively pruning features of unimportant search regions based on attention scores at specific layers. RFGM-Track (Zhou et al., 2023) learns token importance based on attention scores to prune tokens of templates in the memory bank. LMTrack (Xu et al., 2025) prunes reference tokens in memory by aggregating classification scores and attention scores across all layers. Despite significant progress, they commonly perform tracking and inference under fixed token budgets, which pose challenges in flexible computing and deployment scenarios. To overcome this constraint, we integrate token importance estimation with elastic inference enhancement, enabling the tracker to achieve robust performance under elastic token budgets, thereby promoting diverse practical applications.

## 3. Method

In this section, we first introduce the preliminaries of Transformer-based visual trackers. The following subsections present the overview of ETBTrack, token importance estimation and elastic inference enhancement algorithms.

### 3.1. Preliminaries

The core capability of Transformer-based trackers (Ye et al., 2022; Yutao et al., 2022) lies in their feature interaction and extraction through attention mechanisms and MLPs. They first perform patch embedding on both the search region image $I_x \in \mathbb{R}^{3 \times h_x \times w_x}$ and the template image $I_z \in \mathbb{R}^{3 \times h_z \times w_z}$. Formally,

$$T_z^0 = \text{PatchEmbed}(I_z), T_x^0 = \text{PatchEmbed}(I_x). \quad (1)$$

Then, search region tokens $T_x^0 \in \mathbb{R}^{N_x \times C}$ capture cues about the target object through feature interaction with the template tokens $T_z^0 \in \mathbb{R}^{N_z \times C}$ in Transformer layers. Here, $N_z = h_z w_z / p^2$, $N_x = h_x w_x / p^2$. $p$ is the patch size. The basic Transformer layer can be formulated as,

$$[\tilde{T}_z^i; \tilde{T}_x^i] = \text{SelfAttn}([T_z^{i-1}; T_x^{i-1}]) + [T_z^{i-1}; T_x^{i-1}], \quad (2)$$

$$[T_z^i; T_x^i] = \text{MLP}([\tilde{T}_z^i; \tilde{T}_x^i]) + [\tilde{T}_z^i; \tilde{T}_x^i]. \quad (3)$$

Then, the enhanced search region tokens are fed into a box head to regress the target bounding box $b$.

$$b = \text{BoxHead}(T_x^{N_{layer}}), \quad (4)$$

where $N_{layer}$ is the number of Transformer layers. Although some trackers (Zheng et al., 2024; Kang et al., 2025) incorporate temporal propagation modules or integrate additional online templates to model target dynamics, Transformer-based trackers typically retain the fundamental feature interaction and extraction in Equation 2-3. Transformer frameworks provides the flexibility to accommodate inputs with varying numbers of tokens. Some works (Ye et al., 2022; Xu et al., 2025; Zhou et al., 2023) make efforts to prune less important image tokens based on manual criteria for tracking acceleration. However, these trackers are typically constrained to training and inference under fixed token budgets, which still face challenges in flexible computing and deployment scenarios.

### 3.2. Overview of ETBTrack

To achieve robust tracking under elastic token budgets, we propose ETBTrack, a post-training framework comprising two key optimization processes: token importance estimation and elastic inference enhancement, as shown in Figure 2. In the token importance estimation process, we present a novel result-driven importance criteria, in which we optimize a policy network guided by the localization precision of the tracker to rank and select the most critical tokens, thereby directly aligning the objectives of importance estimation and tracking precision. In the elastic inference enhancement process, we develop a new budget-collaborative optimization strategy, in which we collaboratively optimize the tracker across varying budgets by utilizing the most critical tokens identified by the policy network, thereby enabling the tracker to be compatible with diverse budgets.

### 3.3. Token Importance Estimation

The key to achieving robust tracking under elastic token budgets lies in identifying the tokens that are most critical for precise target localization across varying token budgets. Previous trackers (Ye et al., 2022; Xu et al., 2025) typically employ manual criteria (e.g. attention scores) to identify important tokens, which inevitably introduce potential biases and leads to suboptimal results during elastic inference.

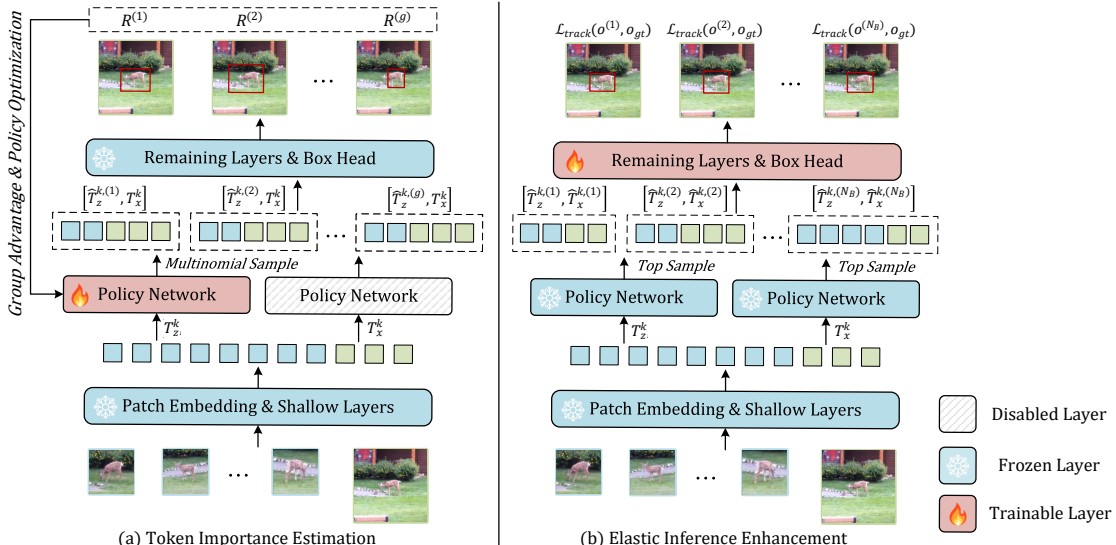

(a) Token Importance Estimation

(b) Elastic Inference Enhancement

*Figure 2.* (a) In the token importance estimation process, we alternately optimize the policy network for the importance estimation of template tokens and search region tokens guided by the localization precision of the tracker, which mitigates potential manual biases for token pruning. (b) In the elastic inference enhancement process, we collaboratively optimize the tracker by utilizing the most critical tokens identified by the policy network across varying budgets, thereby enabling robust tracking performance under elastic token budgets.

As shown in Figure 3(a), using attention scores to prune template or search region tokens may result in imprecise localization results.

To overcome this limitation, we propose a novel result-driven importance criteria to align the objectives of importance estimation and tracking precision. Specifically, we insert two lightweight policy networks after $k^{th}$ Transformer layer for the importance estimation of both template and search region tokens respectively. Given output tokens of $k^{th}$ Transformer layer $T_z^k \in \mathbb{R}^{N_z \times C}$ and $T_x^k \in \mathbb{R}^{N_x \times C}$, the policy network regresses the importance of each token in precisely localizing the target. Formally,

$$P_z = \text{Policy}_z(T_z^k) \in \mathbb{R}^{N_z}, \tag{5}$$

$$P_x = \text{Policy}_x(T_x^k) \in \mathbb{R}^{N_x}. \tag{6}$$

Then, we alternately optimize the policy networks for the template and search region guided by the localization precision of the tracker, as shown in Figure 2(a). Since the policy networks for the template and search region share the same optimization approach, we take the optimization of the template policy network as an example in the following description. Inspired by GRPO (Shao et al., 2024), we sample $g$ group template tokens $\{\hat{T}_z^{k,(i)}\}_{i=1}^g$ using a multinomial distribution parameterized by the importance scores output from the policy network. Formally,

$$p = \text{Softmax}(P_z), \tag{7}$$

$$\mathcal{S}_z^{(i)} \sim \text{Multinomial}\left(N_{sample}, p\right), \tag{8}$$

$$\hat{T}_z^{k,(i)} = T_z^k[\mathcal{S}_z^{(i)}, :] \in \mathbb{R}^{N_{sample} \times C}. \tag{9}$$

where $N_{sample}$ is the number of tokens sampled in each group, $\mathcal{S}_z^{(i)} \subset \mathcal{Z}_z$ denotes the sampled token index set. $\mathcal{Z}_z = \{1, 2, ..., N_z\}$ is the universal set of the template token indexes. The sampled $g$ group template tokens are concatenated with the search region tokens $\{[\hat{T}_z^{k,(i)}, T_x^k]\}_{i=1}^g$ to build interactions in the remaining Transformer layers and predict the target bounding box $\{\hat{b}^{(i)}\}_{i=1}^g$ separately. Then, we employ the IoU between predicted and groundtruth bounding boxes $b_{gt}$ as the reward signal $R$ and compute the advantage function $A$ for these token groups. Formally,

$$R^{(i)} = \text{IoU}(\hat{b}^{(i)}, b_{gt}), \tag{10}$$

$$A^{(i)} = \frac{R^{(i)} - \text{mean}(\{R^{(i)}\}_{i=1}^g)}{\text{std}(\{R^{(i)}\}_{i=1}^g)}, \tag{11}$$

where $\text{mean}(\cdot)$ denotes the mean of the value set, $\text{std}(\cdot)$ denotes the standard deviation of the value set. For ease of exposition, we reformulate the template policy network with parameter $\theta$ as $\pi_\theta^z(T_x^k, \mathcal{S})$. Formally,

$$\pi_\theta^z(j|T_z^k, \mathcal{S}) = \begin{cases} \sigma(P_z), & \text{if } j \in \mathcal{S} \\ 1 - \sigma(P_z), & \text{if } j \in \mathcal{Z}_z \setminus \mathcal{S} \end{cases} \tag{12}$$

where $\sigma(\cdot)$ denotes sigmoid operation. $P_z$ is defined in Equation 5. Similar to GRPO (Shao et al., 2024), we optimize the policy network by maximizing the following

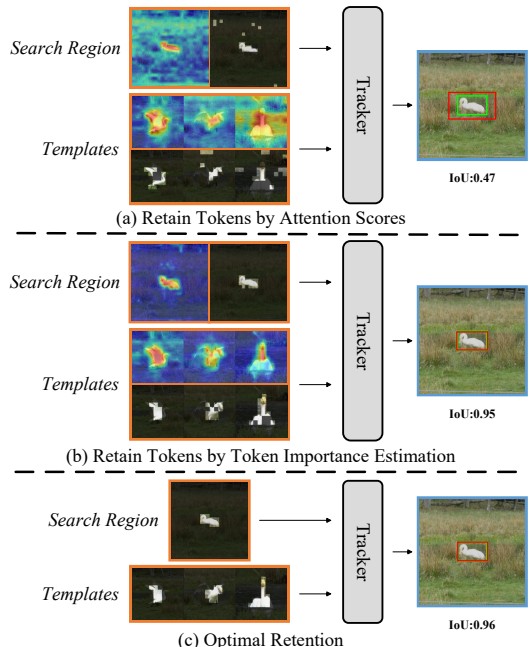

(a) Retain Tokens by Attention Scores

(b) Retain Tokens by Token Importance Estimation

(c) Optimal Retention

*Figure 3.* The optimal retention, obtained via greedy search, corresponds to the most critical token combination for precise target localization. The token combination retained by our token importance estimation closely approximates this optimum, whereas attention-based retention yields a suboptimal result.

objective function,

$$\mathcal{J}^z(\theta) = \mathbb{E}_{j \in \mathcal{Z}_z} \left[ \sum_{i=1}^g r_{i,j}^z(\theta) - \beta \mathbb{D}_{KL}(\pi_\theta^z || \pi_{\theta_{ref}}^z) \right], \tag{13}$$

$$r_{i,j}^z(\theta) = \pi_\theta^z \left( j | T_z^k, \mathcal{S}^{(i)} \right) A^{(i)}. \quad j \in \mathcal{Z}_z \tag{14}$$

Here, $\pi_{\theta_{ref}}^z$ refers to the reference policy, whose parameters are updated less frequently than $\pi_\theta^z$. $\mathbb{D}_{KL}(\cdot || \cdot)$ denotes Kullback-Leibler divergence, a regularization loss to prevent significant deviations from the reference policy during optimization. $\beta$ is the regularization coefficient. The above objective function drives the policy network to optimize tokens with greater precision advantage toward higher importance scores.

### 3.4. Elastic Inference Enhancement

Existing Transformer-based trackers (Xin et al., 2023; Zheng et al., 2024; Ye et al., 2022) perform both inference and training with a fixed number of tokens (full tokens or pruned tokens). Consequently, when tracking with a different token budget than used during training, their performance remains limited even if we retain the optimal tokens. To clearly characterize the capabilities of the tracker under varying token budgets, we define the performance upper bound under token budgets of template $B_z$ and search

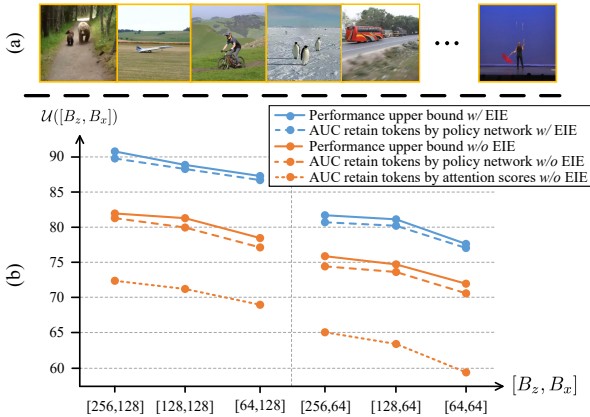

*Figure 4.* (a) We sampled 100 sets of samples to explore the performance upper bound. These samples cover diverse categories and challenging scenarios. (b) We employ greedy search to determine the performance upper bounds of these samples. Token importance estimation effectively approximates these upper bounds, while elastic inference enhancement consistently elevates them.

region $B_x$ as follows,

$$\mathcal{U}([B_z, B_x]) = \max_{\substack{T_z' \subseteq T_z, \ |T_z'|=B_z \\ T_x' \subseteq T_x, \ |T_x'|=B_x}} f([T_z', T_x']), \tag{15}$$

where $T_z \in \mathbb{R}^{N_z \times C}, T_x \in \mathbb{R}^{N_x \times C}$ denotes the universal set of template and search region tokens respectively, $T_z', T_x'$ denotes the sampled token set of template and search region respectively, $f(\cdot)$ denotes the localization precision (IoU) of the tracker with the specific token set. Since sufficient cues for token pruning are only available after the initial token interactions, we typically enable interactions among all tokens within the first $k$ layers and perform token pruning after the $k^{th}$ layer, after which subsequent layers perform interactions using the pruned tokens. For simplicity, in the equations presented in this section, the model input directly adopts the representation of pruned tokens. We explore the performance upper bound of SeqTrack (Xin et al., 2023) under varying token budgets through 100 diverse samples as shown in Figure 4(b). When employing a limited token budget for target localization, the tracker exhibits a limited performance upper bound, which is attributed to the budget discrepancy between training (full tokens) and inference (pruned tokens).

To address the above limitation, we propose an elastic inference enhancement algorithm to improve the performance upper bound of the tracker under varying token budgets. Independently alternating the optimization of tracking capabilities across different token budgets is prone to training instability and capability bias. To address this, we treat the capabilities of the tracker under varying token budgets as a holistic objective for joint optimization, thereby identifying a common optimization direction that enhances capabilities across different token budgets. As shown in Figure 2(b), we

freeze the policy network and all preceding layers, and utilize the trained policy network to select the most important tokens under different budgets $\{B_z^{(i)}, B_x^{(i)}\}_{i=1}^{N_B}$. Formally,

$$\mathcal{I}_z^{(i)} = \text{argtop}(P_z, B_z^{(i)}), \tag{16}$$

$$\mathcal{I}_x^{(i)} = \text{argtop}(P_x, B_x^{(i)}), \tag{17}$$

$$T_z^{(i)} = T_z[\mathcal{I}_z^{(i)}, :] \in \mathbb{R}^{B_z \times C}, \tag{18}$$

$$T_x^{(i)} = T_x[\mathcal{I}_x^{(i)}, :] \in \mathbb{R}^{B_x \times C}. \tag{19}$$

where $P_z \in \mathbb{R}^{N_z}, P_x \in \mathbb{R}^{N_x}$ are importance scores of template and search region tokens as defined in Equation 5,6. $N_B$ denotes the number of budgets used for collaborative optimization. Subsequently, we use these sampled tokens under different budgets for collaborative optimization, thereby enhancing the tracking precision of the tracker under varying token budgets. Formally,

$$o^{(i)} = \mathcal{M}([T_z^{(i)}, T_x^{(i)}]), \tag{20}$$

$$\mathcal{L} = \sum_{i=1}^{N_B} \mathcal{L}_{track}(o^{(i)}, o_{gt}), \tag{21}$$

where $o^{(i)}$ denotes the output of the tracker $\mathcal{M}(\cdot)$. $o_{gt}$ is the corresponding groundtruth of the outputs. The loss $L_{track}$ is consistent with the training loss of the pretrained tracker.

**Discussion**. To achieve elastic token tracking for visual trackers, two critical considerations arise from the perspective of the performance upper bound: (1) Retain the most critical tokens to approach the performance upper bound of the tracker under different token budgets. (2) Raise the performance upper bound of the tracker across varying token budgets. The two optimization processes we designed also closely revolve around these two aspects. The token importance estimation optimization identifies and selects the most critical tokens in precise localization. The elastic inference enhancement optimization elevates the performance upper bound to ensure robustness and adaptability under dynamic resource constraints. As shown in Figure 4(b), our token importance estimation can approximate the performance upper bound of the tracker across varying token budgets, whereas the manual pruning algorithm (attention-based) is constrained in performance. Furthermore, our elastic inference enhancement effectively improves the performance upper bound of the tracker under diverse token budgets. This collectively demonstrates the efficacy of the ETBTrack training framework.

## 4. Experiment

### 4.1. Implementation Details

All experiments are conducted on a server with 8 RTX 3090 GPUs and AMD EPYC 7713 64-Core CPU Processor. The random seed is set to 42.

**Training Details**. Without loss of generality, we validate the ETBTrack post-training framework on two representative Transformer-based visual trackers featuring distinct designs, specifically SeqTrack (Xin et al., 2023) and ODTrack (Zheng et al., 2024). SeqTrack establishes interactions between multiple templates and the search region in the Transformer, with the enhanced search region autoregressively predicting the bounding box using a causal Transformer. ODTrack establishes interactions between the search region and multiple templates in the Transformer while introducing temporal tokens to propagate information across frames, with the enhanced search region regressing the bounding box via a convolutional box head. They represent different temporal modeling approaches and bounding box prediction strategies. We also validate the effectiveness of ETBTrack across Transformers of varying scales and input images with diverse resolutions. The network architecture and training data are identical to those of the pretrained tracker. For trackers employing ViT-large (Dosovitskiy et al., 2021), we insert the policy networks after the fourth layer; for those utilizing ViT-base, the policy network is integrated after the second layer. In the *token importance estimation* process, $N_{sample}$ is set to 0.5 and 2 times the number of target tokens $N_{target}$ for the template and search region, respectively. The number of target tokens can be calculated by multiplying the total number of tokens in the search region with the area ratio between the target size and the search region set during training. We sample 32 group tokens per sample for exploration. The regularization coefficient $\beta = 10$. We employ a batch size of 16 and a learning rate of $1 \times 10^{-4}$ for policy optimization. The training process spans 6 epochs, with each epoch processing 6,400 samples. In the *elastic inference enhancement* process, we set $B_z \in \{N_{target}, 2N_{target}, 4N_{target}, 8N_{target}\}$ and $B_x \in \{4N_{target}, 8N_{target}\}$ for collaborative optimization with different token budgets. We employ a batch size of 32 and a learning rate of $1 \times 10^{-6}$. The training process spans 15 epochs, with each epoch processing 10,000 samples. We alternately perform the above two optimization processes twice. Notably, the post-training process is efficient. The total post-training duration represents only 14.3%–19.2% of the original base model's training time in the same environment.

**Inference Details**. The hyperparameters used for tracking are consistent with those of the pretrained trackers (Xin et al., 2023; Zheng et al., 2024). The only difference is that we perform tracking with the most critical tokens estimated by the policy networks under different token budgets.

### 4.2. Tracking with Elastic Token Budgets

**Comprehensive Results of ETBTrack**. As shown in Figure 5, the inference speed, peak memory consumption, and FLOPs of the tracker exhibit significant variations as the

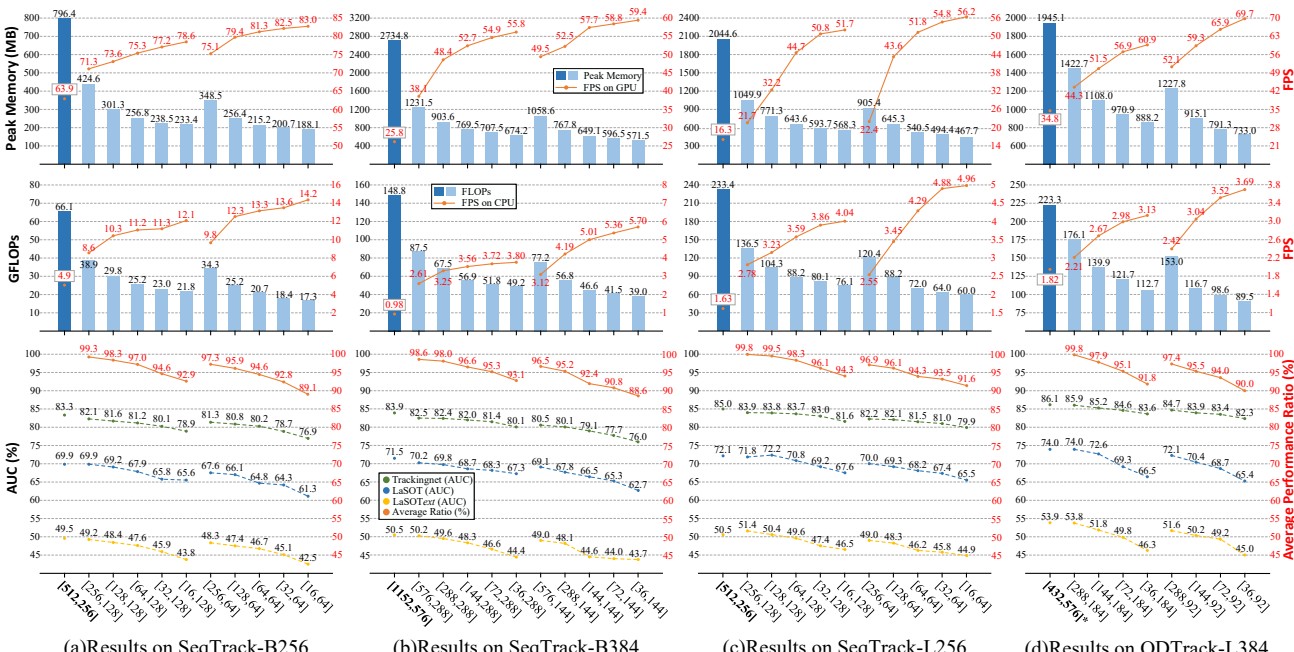

*Figure 5.* The x-axis $[B_z, B_x]$ represents the token budget allocated to the template and search region, respectively. The inference speed, peak memory consumption, and FLOPs of trackers exhibit significant variations as the token budget changes. Trackers trained with ETBTrack demonstrate robust performance with a elastic token budget. $*$ denotes the original tracker trained with manual pruning under a specific token budget. Zooming in for a better view.

token budget changes. This highlights that the token budget plays a critical role in adapting the tracker to different computational environments in practical applications. We evaluate on LaSOT (Fan et al., 2019), LaSOT$_{ext}$ (Fan et al., 2021) and Trackingnet (Muller et al., 2018), and report the main metric (AUC) in Figure 5. ETBTrack empowers trackers with elastic token budgets, enabling a single tracking framework to dynamically adapt to diverse computational requirements across varying deployment scenarios. Notably, SeqTrack-L256 trained using ETBTrack achieves 99.5% average performance with 33.3% tokens, 37.7% peak memory, and 2.0× inference speed. Furthermore, compared to the original ODTrack trained with manual pruning under a specific token budget, training with ETBTrack achieves comparable performance with the token budget of [288,184] while only requiring 78.8% FLOPs and 73.1% peak memory. Moreover, we compare ETBTrack with other state-of-the-art token pruning approaches, as summarized in Table 1. For a fair comparison, pruning is consistently applied after the second layer for all methods. Following their original implementations, OSTrack prunes tokens based on attention scores relative to the template's center token, while LM-Track utilizes the average attention scores across all search region tokens. Both OSTrack* and LMTrack* are trained under a fixed token budget $[B_z, B_x]$ of [256,128]. The results demonstrate that ETBTrack significantly outperforms existing pruning strategies with similar FPS. Notably, ETBTrack further enables robust tracking under elastic token

*Table 1.* Comparison with other token pruning methods using SeqTrack-B256 as base model. * means that we adopt the corresponding pruning strategy.

| method | $FPS_{CPU}$ | $FPS_{GPU}$ | Average Ratio (%) |
|---|---|---|---|
| ETBTrack[256,128] | 8.6 | 71.3 | 99.3 |
| OSTrack*[256,128] | 8.9 | 71.9 | 86.9 |
| LMTrack*[256,128] | 8.8 | 71.7 | 72.4 |

budgets, which greatly facilitates flexible computation and deployment in diverse scenarios.

**Performance Comparison with Other Trackers**. We have conducted a performance comparison with other representative trackers, as shown in Table 2. By comparing ETBTrack with other strong full-budget trackers, it is evident that ETBTrack offers a more flexible Pareto-optimal frontier. While other trackers are constrained to fixed computational budgets, our framework enables a single model to adapt to varying computational demands while consistently delivering competitive tracking results. By dynamically adjusting the token budget, ETBTrack achieves a superior performance-efficiency trade-off, enabling ODTrack to simultaneously outperform other methods in both accuracy and computational efficiency.

### 4.3. Ablation Study

We take the ablation study on SeqTrack-B256 with token budgets $[B_z, B_x]$ of [32,128], [128,128] and [128,64].

*Table 2.* Comparison with state-of-the-art trackers on LaSOT and TrackingNet.

| Method | LaSOT (AUC) | TrackingNet (AUC) | FPS |
|---|---|---|---|
| HIPTrack (Cai et al., 2024) | 72.7 | 84.5 | 42.6 |
| ARTrack-384 (Wei et al., 2023) | 72.6 | 85.1 | 13.2 |
| TATrack-L (Huang et al., 2024) | 71.1 | 85.0 | 7.2 |
| AQATrack-384 (Xie et al., 2024) | 72.7 | 84.8 | 43.8 |
| ROMTrack-384 (Cai et al., 2023) | 71.4 | 84.1 | 28.0 |
| EVPTrack-384 (Shi et al., 2024) | 72.7 | 84.4 | 29.6 |
| ARTrackV2-L384 (Bai et al., 2024) | 73.6 | 86.1 | 43.2 |
| LoRAT-g-224 (Lin et al., 2024) | 74.9 | 85.2 | 41.6 |
| SMTrack-M256 (Ma et al., 2026) | 70.1 | 84.2 | 36.0 |
| SMTrack-M384 (Ma et al., 2026) | 71.9 | 85.2 | 34.0 |
| MCITrack-T224 (Kang et al., 2025) | 71.7 | 84.8 | 50.3 |
| MCITrack-S224 (Kang et al., 2025) | 73.8 | 85.6 | 40.7 |
| ODTrack-L384 (Zheng et al., 2024) | 74.0 | 86.1 | 34.8 |
| ETBTrack(ODTrack-L384) [288, 184] | 74.0 | 85.9 | 44.3 |
| ETBTrack(ODTrack-L384) [288, 92] | 72.1 | 84.7 | 52.1 |
| ETBTrack(ODTrack-L384) [144, 184] | 72.6 | 85.2 | 51.5 |

*Table 3.* Analysis of the sample strategy.

| group size $g$ | sample | [32,128] | [128,128] | [128,64] |
|---|---|---|---|---|
| 16 | multinomial | 62.8 | 66.1 | 63.7 |
| 32 | multinomial | 63.2 | 66.3 | 64.1 |
| 64 | multinomial | 63.2 | 66.3 | 64.2 |
| 32 | random | 61.9 | 64.7 | 62.4 |

*Table 4.* Analysis of the regularization loss.

| $\beta$ | sync step | [32,128] | [128,128] | [128,64] |
|---|---|---|---|---|
| 10.0 | 100 | 63.0 | 66.0 | 63.9 |
| 10.0 | 1,000 | 63.2 | 66.3 | 64.1 |
| 10.0 | 10,000 | 62.7 | 66.2 | 63.5 |
| 1.0 | 1,000 | 61.1 | 64.6 | 61.7 |
| 100.0 | 1,000 | 62.2 | 65.6 | 63.1 |

*Table 5.* Analysis of the policy network position.

| layer | prune | [32,128] | | [128,128] | | [128,64] | |
|---|---|---|---|---|---|---|---|
| | | AUC | GFLOPs | AUC | GFLOPs | AUC | GFLOPs |
| 1 | policy | 62.6 | 18.725 | 65.8 | 26.206 | 63.4 | 21.189 |
| 2 | policy | 63.2 | 23.032 | 66.3 | 29.833 | 64.1 | 25.270 |
| 4 | policy | 63.3 | 31.646 | 66.3 | 37.087 | 64.3 | 33.431 |

*Table 6.* Analysis of the training strategies of elastic inference enhancement.

| training strategy | [32,128] | [128,128] | [128,64] |
|---|---|---|---|
| w/o EIE | 63.2 | 66.3 | 64.1 |
| reduction | 66.2 | 67.8 | 65.7 |
| alternate | 65.0 | 68.5 | 66.0 |
| collaborative | 65.8 | 69.2 | 66.1 |

*Table 7.* Analysis of the alternating training of two optimization processes.

| number of alternations | [32,128] | [128,128] | [128,64] |
|---|---|---|---|
| 1 | 65.4 | 68.8 | 65.6 |
| 2 | 65.8 | 69.2 | 66.1 |
| 3 | 65.9 | 69.2 | 66.2 |

**Analysis of Token Importance Estimation**. The experiment in this analysis only optimizes with token importance estimation. As shown in Table 3, utilizing multinomial sampling for token selection can leverage online learning experiences to explore optimal token combinations, achieving performance gains. Furthermore, increasing the group size can improve performance, albeit with diminishing returns. In the regularization loss, we periodically synchronize the parameters of the reference policy with those of the training policy at fixed intervals of training steps. An appropriate regularization coefficient combined with a suitable synchronization interval can effectively improve performance, as shown in Table 4. Inserting the policy networks at earlier layers reduces computational cost but compromises importance estimation, leading to performance degradation. Empirical results in Table 5 demonstrate that placing them after the second layer achieves an optimal balance between model performance and computational efficiency.

**Analysis of Elastic Inference Enhancement**. With elastic inference enhancement (EIE), the tracker can effectively enhance its performance across varying token budgets, as shown in Table 6. During training, both the gradual reduction of token budgets and the alternating selection of token budgets for optimization result in performance bias in the trained models. Differently, we treat the capabilities of the tracker under varying token budgets as a holistic objective for collaborative optimization, thereby identifying a common optimization direction that enhances capabilities across different token budgets.

**Analysis of Alternating Training**. Since token selection and parameter optimization are interdependent, we need to alternately perform the two optimization procedures multiple times. Fortunately, this alternating process converges rapidly to an optimal solution, as demonstrated in Table 7.

*Table 8.* Comparison with manual pruning criteria.

| train | prune | [32,128] | [128,128] | [128,64] |
|---|---|---|---|---|
| - | Policy | 63.2 | 66.3 | 64.1 |
| - | $Attn_t$ | 46.3 | 48.8 | 45.9 |
| - | $Attn_s$ | 26.6 | 27.3 | 25.9 |
| Policy EIE | Policy | 65.8 | 69.2 | 66.1 |
| $Attn_t$ EIE | $Attn_t$ | 47.2 | 49.3 | 46.2 |
| $Attn_s$ EIE | $Attn_s$ | 28.1 | 28.6 | 29.5 |
| Random EIE | Random | 10.2 | 19.8 | 6.3 |

We can find that the performance of the tracker can converge after two alternations. Further optimization will hardly bring about any performance improvement. Thus, we alternately perform the optimization processes of token importance estimation and elastic inference enhancement twice.

**Comparison with Manual Pruning Criteria**. We further conduct a comparison with popular manual pruning criteria. 1) Pruning by attention scores received from the center token of the template ($Attn_t$). This is adopted in the early candidate elimination of OSTrack (Ye et al., 2022). 2) Pruning by average attention scores received from all the search region tokens ($Attn_s$). This is adopted in the reference token updating of LMTrack (Xu et al., 2025). As shown in Table 8, it is observed that directly employing attention scores as an indicator of token importance for elastic token budgets incurs severe performance degradation under such small token budgets. This stems from the potential manual biases, which fails to identify tokens most critical for precise target localization. In contrast, our proposed policy network is optimized guided by the localization precision of the tracking, aligning the objectives of importance estimation and tracking precision. This enables the trackers to identify the most important token for precise target localization under elastic token budgets. Furthermore, we explored whether other pruning strategies during training could achieve robust tracking under elastic budgets. Results in rows 4-6 of Table 8 indicate that this approach yields negligible gains, primarily due to the discrepancy of objectives between the token pruning and tracking precision. This underscores the necessity of our designed policy network to handle diverse computational constraints in robust tracking.

**More interesting discussion and visualization results** are shown in Appendices.

## 5. Conclusion

We propose a novel post-training framework for visual tracking, ETBTrack, that enables trackers perform robust tracking under elastic token budgets to accommodate diverse computational requirements. We present a novel result-driven importance criteria, in which we optimize a lightweight policy network guided by the localization preci-

sion of the tracker, thereby directly aligning the objectives of importance estimation and tracking precision. We develop a new budget-collaborative optimization strategy to identifying a common optimization direction for various token budgets. Extensive experimental results and ablation study demonstrate the effectiveness of ETBTrack. Beyond visual tracking, ETBTrack offers a promising paradigm for elastic inference that could be generalized to broader domains.

## Impact Statement

This paper presents work whose goal is to advance the field of Machine Learning. There are many potential societal consequences of our work, none which we feel must be specifically highlighted here.

## Acknowledgments

This work was supported by the National Natural Science Foundation of China (U25A20536), Youth Innovation Promotion Association of CAS.

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

*Table 9.* Comparison with manual pruning criteria using SeqTrack-B256 on LaSOT in term of AUC. $Attn_t$ - Pruning by attention scores received from the center token of the template. This is adopted in the early candidate elimination of OSTrack (Ye et al., 2022). $Attn_s$ - Pruning by average attention scores received from all the search region tokens. This is adopted in the reference token updating of LMTrack (Xu et al., 2025).

| pruning | [32,128] | [128,128] | [128,64] |
|---------|----------|-----------|----------|
| *Policy* | 63.2 | 66.3 | 64.1 |
| $Attn_t$ | 46.3 | 48.8 | 45.9 |
| $Attn_s$ | 26.6 | 27.3 | 25.9 |

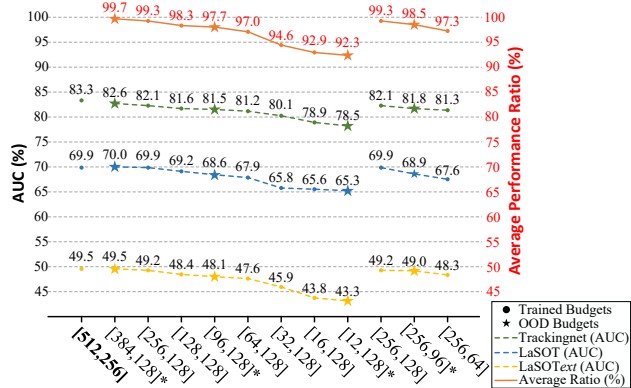

*Figure 6.* Analysis of the generalization of token budgets

# A. Experimental Discussions

## A.1. Evaluation Metric

We perform a one-pass evaluation and report the area under the curve (Wu et al., 2015) of the success plot for LaSOT, LaSOT$_{ext}$ and TrackingNet. The success score is measured as the Intersection over Union (IoU) of the pixels between the groundtruth bounding boxes $b_{gt}$ and the ones generated by the trackers $\hat{b}$. We mark AUC for LaSOT/LaSOT$_{ext}$/TrackingNet in Figure 5 of the main manuscript as habitual abbreviations. All the main experimental results are run three times to ensure their reproducibility and stability.

## A.2. Performance Discussion

The experiment in this analysis adopt the original pretrained model. The token pruning mechanisms in OSTrack (Ye et al., 2022) and LMTrack (Xu et al., 2025) are manually designed for a specific token budget or tailored model architectures. The early candidate elimination strategy in OS-Track progressively prunes a small number of tokens within the search region, which results in limited computational acceleration. For instance, when ODTrack (Zheng et al., 2024) employs early candidate elimination, it reduces the computational load from 311.2 GFLOPs to 223.3 GFLOPs. In contrast, ODTrack trained with ETBTrack achieves similar performance (99.8%) with a token budget $[B_z, B_x]$ of $[288, 184]$ at 176.1 GFLOPs, and further maintains 93.6% of the original performance with merely 98.6 GFLOPs under a token budget of $[72, 92]$. Notably, ETBTrack enables the tracker with elastic token budgets for tracking, whereas early candidate elimination is constrained to operate under a fixed token budget for both training and tracking. When early candidate elimination is applied in shallow layers under constrained token budgets, the performance exhibits a sharp decline, as shown in Table 9 ($attn_t$). LMTrack leverages the aggregation of average attention scores received from the search region tokens and the classification scores to prune tokens in memory, which is meticulously engineered for the specific memory mechanism and network designs that inherently require the output of classification

scores. Directly pruning tokens by the average attention scores received from the search region tokens for trackers without classification scores (*e.g.* SeqTrack) invariably leads to significant performance degradation, as shown in Table 9 ($attn_s$). Unlike the above manual importance criteria based on attention scores, our ETBTrack directly optimizes a lightweight policy network guided by the localization precision of the tracker, thereby mitigating the potential biases introduce by the manual criteria and ensuring seamless compatibility with Transformer-based trackers. As shown in Table 10, ETBTrack enables trackers to perform robust tracking over a wide range of computational capabilities.

## A.3. Budget Generalization

To evaluate the generalization capability of our model, we conduct tracking using token budgets that were not encountered during training. As illustrated in Figure 6, ETBTrack maintains robust tracking performance even when operating under unseen token budgets. Notably, the performance remains stable compared to adjacent budget settings. These results demonstrate the superior out-of-distribution generalization of ETBTrack across varying computational budgets.

## A.4. Analysis of the Policy Network Costs

As shown in Table 11, the policy network accounts for only a negligible fraction of the total parameters and computational FLOPs. This demonstrates that the network can achieve result-driven importance estimation for tokens with minimal overhead.

## A.5. Analysis of Collaborative Training

We study the token budgets we used for collaborative training. As shown in Table 12, When optimizing parameters under a fixed token budget, the performance under the corresponding budget improve slightly, while the performance under other token budgets degrade significantly. This in-

*Table 10.* The dynamic range compared to the original trackers under elastic token budgets. ETBTrack enables trackers to perform robust tracking over a wide range of computational capabilities. ∗ denotes the original tracker trained with manual pruning under a specific token budget.

| | Peak Memory ↓ | GFLOPs ↓ | $FPS_{GPU}$ ↑ | $FPS_{CPU}$ ↑ | Avg. AUC ↑ |
|---|---|---|---|---|---|
| SeqTrack-B256 | 23.6%∼53.3% | 26.2%∼58.9% | 1.3×∼1.1× | 2.9×∼1.8× | 88.6%∼99.3% |
| SeqTrack-B384 | 20.9%∼45.0% | 26.2%∼58.8% | 2.3×∼1.5× | 5.8×∼2.7× | 88.3%∼98.6% |
| SeqTrack-L256 | 22.9%∼51.3% | 25.7%∼58.5% | 3.4×∼1.3× | 3.0×∼1.7× | 91.3%∼100.0% |
| ODTrack-L384* | 37.7%∼73.1% | 40.1%∼78.9% | 2.0×∼1.3× | 2.0×∼1.2× | 89.2%∼99.8% |

*Table 11.* Analysis of the policy network

| method | Params (M) | GFLOPs |
|---|---|---|
| SeqTrack-B256 | 89.1 | 66.1∼17.3 |
| polict network | 0.78 | 0.30 |
| SeqTrack-B384 | 89.1 | 148.8∼39.0 |
| polict network | 0.78 | 0.68 |
| SeqTrack-L256 | 306.5 | 233.4∼60.0 |
| polict network | 1.40 | 0.53 |
| ODTrack-L384 | 311.3 | 223.3∼89.5 |
| polict network | 1.40 | 0.70 |

*Table 12.* Analysis of the token budgets used for elastic inference enhancement.

| token budgets | [32,128] | [128,128] | [128,64] |
|---|---|---|---|
| fixed [32,128] | 65.9 | 66.3 | 65.5 |
| fixed [128,128] | 61.3 | 69.4 | 63.7 |
| fixed [128,64] | 62.7 | 67.1 | 66.4 |
| collaborative | 65.8 | 69.2 | 66.1 |

dicates the necessity of collaborative training for elastic inference.

### A.6. Analysis of the Sample Strategy during Training

During training, as shown in Table 13, sampling fewer tokens enables the tracker to identify the more critical tokens, thereby improving performance. However, excessively reducing the token count may lead to unstable reward signals during training, thereby degrading performance.

### A.7. Experiments on Mobile Devices

Since elastic token budgets are particularly relevant for resource-constrained platforms, we conduct extensive efficiency measurements on the NVIDIA Jetson Orin NX (16GB RAM), which is a representative mobile-computing platform. The hardware configuration includes a 1024-core NVIDIA Ampere Architecture GPU and an ARM Cortex-A78AE CPU. As shown in Table 14, ETBTrack allows for significant adjustments to tracking efficiency (up to 2.9 × FPS) on mobile devices by varying the token budget. These results validate the capability of our proposed ETBTrack to meet the diverse computational requirements of mobile devices.

*Table 13.* Analysis of the number of tokens sampled in each group during training. The AUC on LaSOT is reported, similarly hereinafter.

| $N_{sample}$ | | [32,128] | [128,128] | [128,64] |
|---|---|---|---|---|
| template | search | | | |
| $0.25N_{target}$ | $1.0N_{target}$ | 63.0 | 65.9 | 63.5 |
| $0.5N_{target}$ | $2.0N_{target}$ | 63.2 | 66.3 | 64.1 |
| $1.0N_{target}$ | $4.0N_{target}$ | 62.4 | 65.6 | 62.8 |

*Table 14.* Experiments on mobile devices under different token budgets. $[B_z, B_x]$ denotes the token budget for the template and search region, respectively.

| Tracker | Token Budget | CPU FPS | GPU FPS |
|---|---|---|---|
| SeqTrack-B256 | [512, 256] | 0.73 | 4.29 |
| | [256, 128] | 1.30 | 7.41 |
| | [128, 128] | 1.58 | 8.77 |
| | [64, 128] | 1.75 | 9.80 |
| | [32, 128] | 1.83 | 10.20 |
| | [16, 128] | 1.85 | 10.42 |
| ETBTrack(SeqTrack-B256) | [256, 64] | 1.47 | 8.33 |
| | [128, 64] | 1.78 | 9.71 |
| | [64, 64] | 1.86 | 10.53 |
| | [32, 64] | 2.11 | 10.99 |
| | [16, 64] | 2.23 | 11.24 |

### A.8. Comparison with Other Token Pruning Methods

We have conducted a systematic performance comparison between our ETBTrack and various state-of-the-art token pruning methods. We implemented existing pruning strategies (based on the SeqTrack-B256) following their original designs and compared them with our approach under different token configurations $[B_z, B_x]$. As shown in Table 15, ETBTrack consistently achieves superior performance across all budget settings, demonstrating its effectiveness in maintaining high accuracy under elastic constraints. Unlike training-free token reduction, which causes significant performance degradation, our policy network is optimized to precisely select tokens for robust tracking. This tailored approach allows the tracker to maintain robust performance across varying token budgets.

**VisionZip** (Yang et al., 2025a). VisionZip prunes the image tokens into dominant tokens and contextual tokens. We calculate the average attention score that each token receives

*Table 15.* Comparison with various state-of-the-art token pruning methods on SeqTrack-B256. $[B_z, B_x]$ denotes the token budget for the template and search region, respectively. The AUC on LaSOT is reported. Bold indicates the best performance.

| SeqTrack-B256 | [32, 128] | [128, 128] | [128, 64] | Average |
|---|---|---|---|---|
| **ETBTrack (Ours)** | **65.8** | **69.2** | **66.1** | **67.03** |
| VisionZip | 55.8 | 57.6 | 56.4 | 56.60 |
| EViT | 54.7 | 57.4 | 56.7 | 56.27 |
| ToMe | 54.9 | 57.1 | 55.3 | 55.77 |
| OSTrack | 47.2 | 49.3 | 46.2 | 47.57 |
| FastV | 31.5 | 35.3 | 32.9 | 33.23 |
| LMTrack | 28.1 | 28.6 | 29.5 | 28.73 |
| DivPrune | 9.3 | 12.4 | 10.2 | 10.63 |

*Table 16.* Comparison between distillation models and ETBTrack. $[B_z, B_x]$ denotes the token budget for the template and search region, respectively.

| Method | Total Token Number | LaSOT (AUC) |
|---|---|---|
| Distillation-176 | 363 | 66.9 |
| ETBTrack [235,128] | 363 | 69.3 |
| Distillation-144 | 243 | 64.6 |
| ETBTrack [147,96] | 243 | 68.4 |
| Distillation-112 | 147 | 61.9 |
| ETBTrack [51,96] | 147 | 65.3 |
| Distillation-80 | 75 | 54.8 |
| ETBTrack [15,60] | 75 | 60.3 |

from all other tokens, treating those with higher scores as dominant tokens. Subsequently, we spatially sample target tokens uniformly from the remaining non-dominant tokens, while the rest are designated as merge tokens. Based on the cosine similarity between the keys of merge tokens and those of target tokens, each merge token is fused with its most similar target token to form contextual tokens. We retain dominant and contextual tokens at a ratio of 54:10, following the original configuration.

**ToMe** (Bolya et al., 2022). First, the image tokens are evenly divided into two sequences. We then calculate the similarity between the keys of each token across the two sequences and merge the *top-r* most similar token pairs. We iteratively perform four merge operations, reducing the number of image tokens by $4r$ to achieve the desired budgets.

**FastV** (Chen et al., 2024). We compute the average attention-score for one token received from all other tokens as the importance criteria and prune unimportant tokens to the specific budget according to the criteria.

**DivPrune** (Alvar et al., 2025). The DivPrune strategy aims to maintain token diversity by maximizing pairwise distances between the retained tokens. DivPrune defines the distance between two tokens as 1-cos_similarity, implying that a greater distance corresponds to lower similarity. The algorithm maintains two sets: a "selected subset" (initialized as empty) and a "candidate subset" (initialized with all image tokens). The process begins by identifying the token in the candidate subset that possesses the largest "minimum distance" to all other tokens. This token is selected and moved from the candidate subset to the selected subset. In subsequent iterations, for each remaining candidate token, we compute its distance to all tokens already in the selected subset and record the minimum value among these distances. We then select the candidate token associated with the maximum of these recorded minimum distances and incorporate it into the selected subset. This iterative process continues until the desired number of tokens is reached.

## A.9. Comparison With Knowledge Distillation

We have conducted additional experiments to compare ETB-Track with a baseline model trained via standard distillation on low-resolution inputs, including both feature distillation and score map distillation. As shown in Table 16, the proposed ETBTrack significantly outperforms the low-resolution distillation strategy, which clearly demonstrates the effectiveness of ETBTrack.

## A.10. Why GRPO-like Optimization

We employ a GRPO-like optimization strategy to train our policy network, rather than relying on direct supervision, for two primary reasons. 1) **ground-truth labels for absolute importance are inherently difficult to obtain**. Identifying the most critical tokens under various token budgets is non-trivial, as such importance can only be estimated by evaluating the tracking performance across an exhaustive combination of token subsets. Even with the availability of target segmentation masks, a greedy ranking of tokens remains infeasible because the importance scores for most tokens are binary (i.e., 0 or 1), failing to provide a fine-grained ranking signal. 2) **relative importance is sufficient for our objectives**. In practice, ranking tokens based on their relative significance is enough for effective selection, rendering the definition of an absolute importance score unnecessary. Consequently, drawing inspiration from Group Relative Policy Optimization (GRPO), we optimize the policy network by leveraging the relative localization precision (the advantage) among different token combinations. This implies that the scores predicted by the policy network represent relative importance, where their absolute magnitudes hold no intrinsic physical meaning. Naturally, during inference under different token budgets, we can directly preserve the most critical tokens estimated by the policy network for precise localization.

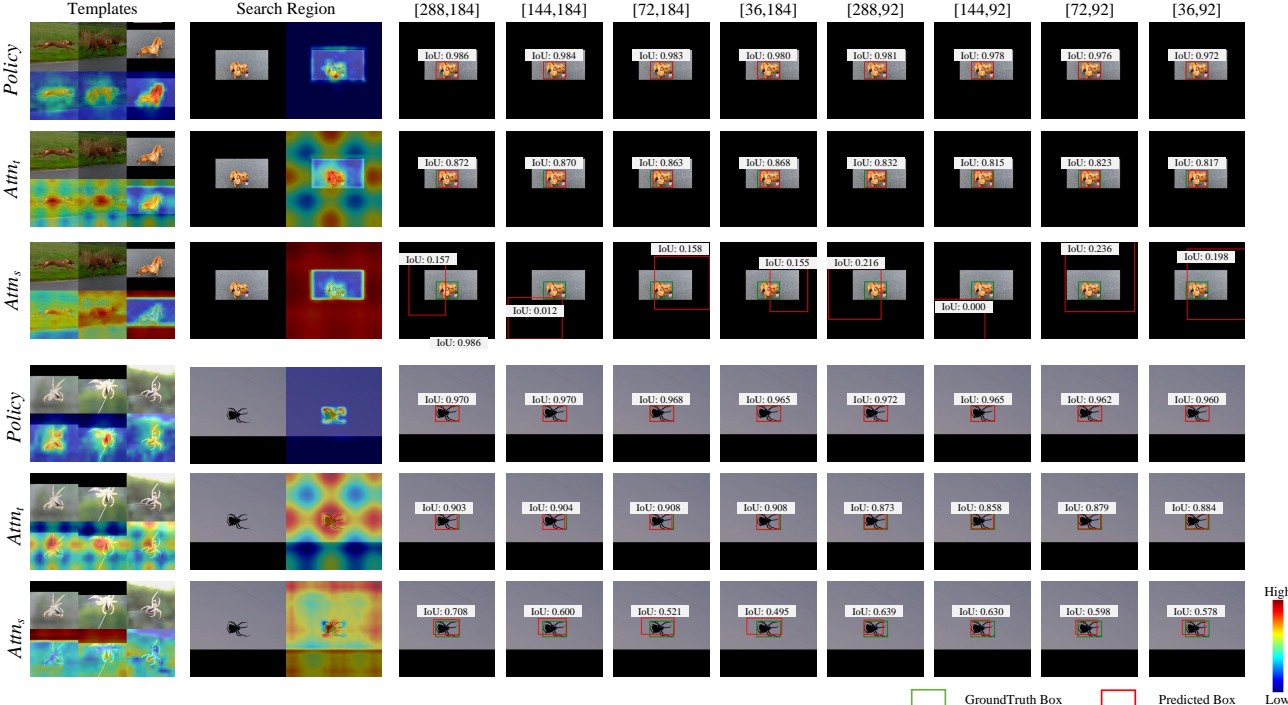

Figure 7. We visualize the importance scores estimated by our policy network or attention scores on ODTrack. $Attn_t$ and $Attn_s$ are defined in Table 9. We can intuitively find that the importance scores estimated by the policy network exhibit enhanced interpretability, with higher scores concentrated in the target region. This enables precise target localization across diverse token budgets.

## B. Visualization

We further visualize additional results of ODTrack with elastic token budgets achieved by pruning tokens using our policy network or manual criteria (attention scores), as shown in Figure 7. Our policy network retains tokens critical for precise target localization through importance score estimation, whereas tokens retained based on attention scores yield suboptimal localization results, thereby compromising tracking robustness. This is because our policy network is directly optimized guided by the localization precision of the tracker, which mitigates the potential biases introduced by the manual criteria. Consequently, the tracker prioritizes the retention of the most critical tokens for precise target localization under diverse computational conditions.

## C. Limitations and Future Work

To ensure training stability, different optimization objectives are currently decoupled, necessitating a two-stage training pipeline. Future work could focus on enhancing optimization efficiency to further streamline the training pipeline. ETBTrack emphasizes elastic token pruning for flexible computation and deployment. The synergistic integration of token pruning and network pruning represents a promising avenue for future exploration.

