# OpenReview forum: "Learning Generalized Trackers with Elastic Token Budgets"
_ICML.cc/2026/Conference — ICML 2026 regular_

### Official Review · Reviewer_nnP4 · 2026-02-24

**Soundness:** 3
**Presentation:** 3
**Significance:** 3
**Originality:** 3
**Overall Recommendation:** 4
**Confidence:** 3

**Summary:**

the paper introduce ETBTrack which improves the tracker efficiency by removing less important token. ETBtrack achieve strong performance on large-scale benchmarks.

**Compliance With Llm Reviewing Policy:**

Affirmed.

**Final Justification:**

See rebuttal response.

**Key Questions For Authors:**

see weakness

**Strengths And Weaknesses:**

strength
- clear motivations
- paper well written and easy to follow

weakness
- the method requires retraining, which incur additional cost
- lack of comparison with more standard methods [1,2,3,4,5,6], but mainly with the original models
- lack of compairson with training-free token reduction approach, such as random token pruning or other more advanced token reduction methods (ToMe [6], Fast-V [7]...etc).

I am willing to adjust the rating if concerns are addressed.

1. Lin, Liting, et al. "Tracking meets lora: Faster training, larger model, stronger performance." European Conference on Computer Vision. Cham: Springer Nature Switzerland, 2024.
2. Xie, Jinxia, et al. "Autoregressive queries for adaptive tracking with spatio-temporal transformers." Proceedings of the IEEE/CVF conference on computer vision and pattern recognition. 2024.
3. Cai, Wenrui, Qingjie Liu, and Yunhong Wang. "Hiptrack: Visual tracking with historical prompts." Proceedings of the IEEE/CVF Conference on Computer Vision and Pattern Recognition. 2024.
4. Zheng, Yaozong, et al. "Odtrack: Online dense temporal token learning for visual tracking." Proceedings of the AAAI conference on artificial intelligence. Vol. 38. No. 7. 2024.
5. Shi, Liangtao, et al. "Explicit visual prompts for visual object tracking." Proceedings of the AAAI conference on artificial intelligence. Vol. 38. No. 5. 2024.
6. Cai, Yidong, et al. "Robust object modeling for visual tracking." Proceedings of the IEEE/CVF international conference on computer vision. 2023.
7. Bolya, Daniel, et al. "Token Merging: Your ViT But Faster." The Eleventh International Conference on Learning Representations.
8. Chen, Liang, et al. "An image is worth 1/2 tokens after layer 2: Plug-and-play inference acceleration for large vision-language models." European Conference on Computer Vision. Cham: Springer Nature Switzerland, 2024.

---

> ### Author Rebuttal · Authors · 2026-03-31
>
> **To Q1**
>
> Compared to other training-free token reduction methods, ETBTrack demonstrates a significant performance advantage. Please refer to Q3 for further details.
>
> **To Q2**
>
> We have included a comparison with more standard methods, including the six methods you suggested, as shown in the table below.
>
> | Method | LaSOT (AUC) | TrackingNet (AUC) | FPS |
> | :--- | :---: | :---: | :---: |
> | HIPTrack | 72.7 | 84.5 | 42.6 |
> | ARTrack-384 | 72.6 | 85.1 | 13.2 |
> | TATrack-L | 71.1 | 85.0 | 7.2 |
> | LoRAT-g-224 | 74.9 | 85.2 | 41.6 |
> | AQATrack-384 | 72.7 | 84.8 | 43.8 |
> | ROMTrack-384 | 71.4 | 84.1 | 28.0 |
> | EVPTrack-384 | 72.7 | 84.4 | 29.6 |
> | ODTrack-L384 | 74.0 | 86.1 | 34.8 |
> | ETBTrack(ODTrack-L384) [288, 184] | 74.0 | 85.9 | 44.3 |
> | ETBTrack(ODTrack-L384) [288, 92] | 72.1 | 84.7 | 52.1 |
> | ETBTrack(ODTrack-L384) [144, 184] | 72.6 | 85.2 | 51.5 |
> *(The notation $[B_z, B_x]$ denotes the elastic token budgets for templates and search regions, respectively.)*
>
> By comparing ETBTrack with other standard trackers, it is evident that ETBTrack offers a more flexible Pareto-optimal frontier. While other trackers are constrained to fixed computational budgets, our framework enables a single model to adapt to varying computational demands while consistently delivering competitive tracking results. By dynamically adjusting the token budget, ETBTrack achieves a superior performance-efficiency trade-off, enabling ODTrack to maintain competitive accuracy with higher inference speed compared to other trackers.
>
> **To Q3**
>
> We have conducted a systematic performance comparison between our ETBTrack and various advanced token reduction methods, including the two methods you suggested.
>
> We implemented existing pruning strategies (based on the SeqTrack-B256) following their original designs and compared them with our approach under different token configurations $[B_z, B_x]$. As shown in the table below, **ETBTrack consistently achieves superior performance across all budget settings**, demonstrating its effectiveness in maintaining high accuracy under elastic constraints.
>
> We can observe that these training-free token reduction strategies lead to significant performance degradation, indicating their incompatibility with the elastic inference requirements of tracking tasks. In contrast, by exploring the localization precision of various token combinations, we optimize a policy network that is well-tailored for token selection in tracking. This enables the tracker to maintain robust performance across varying token budgets. Notably, base trackers trained via ETBTrack achieve robust performance across varying token budgets, which significantly expands its applicability across diverse deployment scenarios.
>
>
> | SeqTrack-B256 | [32, 128] | [128, 128] | [128, 64] | **Average** |
> | :--- | :---: | :---: | :---: | :---: |
> | **ETBTrack (Ours)** | **65.8** | **69.2** | **66.1** | **67.03** |
> | VisionZip | 55.8 | 57.6 | 56.4 | 56.60 |
> | ToMe | 54.9 | 57.1 | 55.3 | 55.77 |
> | FastV | 31.5 | 35.3 | 32.9 | 33.23 |
> | DivPrune | 9.3 | 12.4 | 10.2 | 10.63 |
>
> *Note: $[B_z, B_x]$ denotes the token budget for the template and search region, respectively. Bold indicates the best performance.*
>
> All of the above discussions will be included in the manuscript.

---

> > ### Author Rebuttal · Reviewer_nnP4 · 2026-04-01
> >
> > Thank the authores for the detailed rebuttal and additional discussion on other token pruning method including visionzip, tome, fastv, and divprune. I have several additional follow-up questions:
> >
> > 1. How are these methods implemented? Can you also provide a brief introduction on the implement setup?
> >
> > 2. For Fast-V, they implement the token reduction in the LLM, where do the authores implement Fast-V in SeqTrack without the LLM?
> >
> > 3. Looks like DivPrune achieve the lowest results, can the authors provide more explanation and discussion?
> >
> >
> > ---
> >
> > Thanks the author for the follow-up and additional clairifcation. I will increase my rating to 4.

---

> > > ### Author Response · Authors · 2026-04-02
> > >
> > > We would like to express our sincere gratitude to the reviewer for the insightful comments and the constructive suggestions regarding our token pruning comparison. We are pleased to provide additional details on our implementation setup and further discussion regarding the performance of these methods, specifically addressing the questions concerning FastV and DivPrune.
> > >
> > > ### To Q1
> > >
> > > To ensure a fair comparison, we apply the advanced token reduction algorithm after the second encoder layer of SeqTrack-B256 and separately prune the search region and template image tokens to the specified budgets, following the protocol established in ETBTrack. All implementations strictly follow the original open-source code to ensure a fair comparison.
> > >
> > > **VisionZip**[1]. VisionZip prunes image tokens into dominant tokens and contextual tokens. We calculate the average attention score that each token receives from all other tokens, treating those with higher scores as dominant tokens. Subsequently, we spatially sample target tokens uniformly from the remaining non-dominant tokens, while the rest are designated as merge tokens. Based on the cosine similarity between the keys of merge tokens and those of target tokens, each merge token is fused with its most similar target token to form contextual tokens. We retains dominant and contextual tokens at a ratio of 54:10, following the original configuration.
> > >
> > > **ToMe**[2]. First, the image tokens are evenly divided into two sequences. We then calculate the similarity between the keys of each token across the two sequences and merge the top-r most similar token pairs. We iteratively perform four merge operations, reducing the number of image tokens by 4r to achieve the desired budgets.
> > >
> > > **FastV**[3]. We compute the average attention-score one token received from all other tokens as the importance criteria and prune unimportant tokens to the specific budget according to the criteria.
> > >
> > > **DivPrune**[4]. The DivPrune strategy aims to maintain token diversity by maximizing the pairwise distances between the retained tokens. DivPrune define the distance between two tokens as 1-cos_similarity, implying that a greater distance corresponds to lower similarity. The algorithm maintains two sets: a 'selected subset' (initialized as empty) and a 'candidate subset' (initialized with all image tokens). The process begins by identifying the token in the candidate subset that possesses the largest 'minimum distance' to all other tokens. This token is selected and moved from the candidate subset to the selected subset. In subsequent iterations, for each remaining candidate token, we compute its distance to all tokens already in the selected subset and record the minimum value among these distances. We then select the candidate token associated with the maximum of these recorded minimum distances and incorporate it into the selected subset. This iterative process continues until the desired number of tokens is reached.
> > >
> > > [1] Visionzip: Longer is better but not necessary in vision language models.
> > >
> > > [2] Token Merging: Your ViT But Faster.
> > >
> > > [3] An image is worth 1/2 tokens after layer 2: Plug-and-play inference acceleration for large vision-language models.
> > >
> > > [4] Divprune: Diversity-based visual token pruning for large multimodal models.
> > >
> > > ### To Q2
> > >
> > > FastV utilizes the average attention-score one token received from all other tokens as the importance criteria and prune unimportant tokens to the specific budget according to the criteria. Consistent with this design, we integrate FastV after the second encoder layer of SeqTrack and utilize the attention-score in the second encoder layer as the importance criteria. This pruning position is aligned with that of ETBTrack to ensure a fair comparison.
> > >
> > > ### To Q3
> > >
> > > The focus of DivPrune is to retain diverse image tokens. However, as the background dominates a larger spatial extent in both template and search regions, retaining diverse tokens inevitably results in the preservation of more background-related tokens. The inclusion of diverse background tokens often distracts the tracker and leads to confusion between the target object and the background, resulting in a substantial degradation in performance. This result intuitively suggests that token pruning criteria should be explicitly aligned with the tracking objective to ensure robust performance under varying token budgets.
> > >
> > > Other training-free token reduction methods also utilize manually designed criteria for token pruning. Nevertheless, these heuristics are not inherently aligned with the tracking objective. In contrast, ETBTrack optimizes its policy network directly guided by the localization precision of the tracker. By establishing a direct alignment between token pruning and robust tracking, ETBTrack achieves significant performance advantages.
> > >
> > >
> > > We hope these responses clarify the implementation details and addresses your questions. We remain at your disposal for any further questions or discussions.

---

### Official Review · Reviewer_w21s · 2026-03-02

**Soundness:** 2
**Presentation:** 1
**Significance:** 2
**Originality:** 2
**Overall Recommendation:** 3
**Confidence:** 5

**Summary:**

This paper proposes ETBTrack, which enables trackers perform visual tracking under elastic token budgets to accommodate different computational requirements. ETBTrack uses results-driven importance criteria and a budget-collaborative optimization strategy to identify a common optimization direction for various token budgets. The experimental results show that ETBTrack achieves good tracking results.

**Compliance With Llm Reviewing Policy:**

Affirmed.

**Final Justification:**

While the authors’ rebuttal has addressed some of my concerns, concerns remain regarding SOTA comparisons and the generalization ability of the method. Therefore, I maintain my original rating.

**Key Questions For Authors:**

The authors should restructure the method and polish the writing to help readers better understand the paper’s core content.

**Limitations:**

yes

**Strengths And Weaknesses:**

Strengths
1. ETBTrack enables trackers perform visual tracking under elastic token budgets to accommodate different computational requirements.
2. ETBTrack achieves good tracking results with less computation cost.

Weaknesses
1. The paper is hard to follow. State-of-the-art experiments use line charts to represent the increased reading cost for readers. The framework is relatively simple.
2. The policy layer has to be inserted at a specific Transformer layer, which gives rise to concerns about the method’s generalizability. It is recommended that the authors conduct a thorough analysis of this issue.
3. The different tokens arise from the distinct resolutions of the template and search region fed into the model. It is difficult to discern the benefits of the proposed method from Figure 5. A straightforward reduction in resolution followed by model distillation may yield similar performance.

---

> ### Author Rebuttal · Authors · 2026-03-31
>
> **To Q1**
>
> Figure 5 in the manuscript illustrates key performance metrics, including tracking accuracy, computational complexity, memory consumption, and tracking speed across various token budgets. These results demonstrate that ETBTrack enables the base tracker to achieve robust tracking performance under diverse computational constraints.
>
> To provide a more intuitive comparison with advanced methods, we have included an additional comparison table.
>
> | Tracker | LaSOT (AUC) | TrackingNet (AUC) | FPS |
> | :--- | :---: | :---: | :---: |
> | HIPTrack | 72.7 | 84.5 | 42.6 |
> | ARTrack-384 | 72.6 | 85.1 | 13.2 |
> | TATrack-L | 71.1 | 85.0 | 7.2 |
> | AQATrack-384 | 72.7 | 84.8 | 43.8 |
> | ROMTrack-384 | 71.4 | 84.1 | 28.0 |
> | EVPTrack-384 | 72.7 | 84.4 | 29.6 |
> | **ODTrack-L384** | **74.0** | **86.1** | 34.8 |
> | **ETBTrack(ODTrack-L384) [288, 184]** | **74.0** | **85.9** | **44.3** |
> | **ETBTrack(ODTrack-L384) [288, 92]** | 72.1 | 84.7 | **52.1** |
> | **ETBTrack(ODTrack-L384) [144, 184]** | 72.6 | 85.2 | **51.5** |
> *(The notation $[B_z, B_x]$ denotes the elastic token budgets for templates and search regions, respectively.)*
>
> 1. **Efficiency Gain without Compromising Accuracy:** As demonstrated in the table, `ETBTrack_ODTrack [288, 184]` maintains the high accuracy of the original ODTrack (74.0% vs. 74.0% on LaSOT, 85.9% vs. 86.1% on TrackingNet) while providing a significant boost in inference speed (from 34.8 to 44.3 FPS). This confirms our claim that ETBTrack effectively eliminates redundant tokens with little tracking precision sacrifice.
> 2. **Superior Trade-off:** By comparing ETBTrack with other strong full-budget trackers, it is evident that ETBTrack offers a more flexible Pareto-optimal frontier. While other trackers are constrained to fixed computational budgets, our framework enables a single model to adapt to varying computational demands while consistently delivering competitive tracking results. By dynamically adjusting the token budget, ETBTrack achieves a superior performance-efficiency trade-off, enabling ODTrack to simultaneously outperform other methods in **both accuracy and computational efficiency**.
>
> Further, we have conducted a systematic performance comparison between our ETBTrack and various advanced token reduction methods.
> We implemented existing pruning strategies (based on the SeqTrack-B256) following their original designs and compared them with our approach under different token configurations $[B_z, B_x]$. As shown in the table below, **ETBTrack consistently achieves superior performance across all budget settings**, demonstrating its effectiveness in maintaining high accuracy under elastic constraints. Unlike training-free token reduction, which causes significant performance degradation, our policy network is optimized to precisely select tokens for robust tracking. This tailored approach allows the tracker to maintain robust performance across varying token budgets.
>
>
> | SeqTrack-B256 | [32, 128] | [128, 128] | [128, 64] | **Average** |
> | :--- | :---: | :---: | :---: | :---: |
> | **ETBTrack (Ours)** | **65.8** | **69.2** | **66.1** | **67.03** |
> | VisionZip | 55.8 | 57.6 | 56.4 | 56.60 |
> | ToMe | 54.9 | 57.1 | 55.3 | 55.77 |
> | FastV | 31.5 | 35.3 | 32.9 | 33.23 |
> | DivPrune | 9.3 | 12.4 | 10.2 | 10.63 |
>
> The results presented in the tables above provide an intuitive demonstration of the superiority of our proposed method.
>
> **To Q2**
>
> We thank the reviewer for this insightful comment. We have empirically investigated the impact of the policy layer’s insertion point in the ablation study (Table 4). Our results indicate that while performance varies slightly across layers, the chosen position strikes the optimal balance between tracking accuracy and computational efficiency. Furthermore, the generalizability of ETBTrack is validated by our extensive experiments across various tracking frameworks, input resolutions, and backbone capacities. These results consistently show that our method effectively empowers base trackers with robust performance under elastic token budgets, confirming its efficacy across diverse configurations.
>
> **To Q3**
>
> To address your concern, we have conducted additional experiments to compare ETBTrack with a baseline model trained via standard distillation on low-resolution inputs, including both feature distillation and score map distillation.
>
> | Method | Total Token Number | LaSOT (AUC) |
> | :--- | :---: | :---: |
> | Distillation-176 | 363 | 66.9 |
> | ETBTrack [235,128] | 363 | 69.3 |
> | Distillation-144 | 243 | 64.6 |
> | ETBTrack [147,96] | 243 | 68.4 |
> | Distillation-112 | 147 | 61.9 |
> | ETBTrack [51,96] | 147 | 65.3 |
> | Distillation-80 | 75 | 54.8 |
> | ETBTrack [15,60] | 75 | 60.3 |
> *(The notation $[B_z, B_x]$ denotes the elastic token budgets for templates and search regions, respectively.)*
>
>
> As shown in the table above, the proposed ETBTrack significantly outperforms the low-resolution distillation strategy, which clearly demonstrates the effectiveness of ETBTrack.

---

> > ### Author Rebuttal · Reviewer_w21s · 2026-04-02
> >
> > The authors' response partially addresses my concerns.
> > （1）The experimental comparison remains insufficient; the authors should include more recent, high-performance, low-latency trackers (e.g., MCITrack) to prove a superior performance.
> > (2) The explanation for the policy layer's insertion remains purely empirical; a principled discussion on how it scales across different backbone capacities is needed to justify its generalizability.
> > (3) The manuscript requires significant revision: the core framework diagram and writing are confusing.

---

> > > ### Author Response · Authors · 2026-04-02
> > >
> > > Thank you for your constructive comments and the opportunity to clarify our design.
> > >
> > > ### To Q1
> > > We will include a detailed comparison with recent high-performance, low-latency trackers, including MCITrack, in our final manuscript. The results are summarized in the table below.
> > >
> > > | Tracker | LaSOT (AUC) | TrackingNet (AUC) | FPS |
> > > | :--- | :---: | :---: | :---: |
> > > | SMTrack-M256 | 70.1 | 84.2 | 36.0 |
> > > | MCITrack-T224 | 71.7 | 84.8 | 50.3 |
> > > | ETBTrack(ODTrack-L384) [144,184] | **72.6** | **85.2** | **51.5** |
> > > | SMTrack-M384 | 71.9 | 85.2 | 34.0 |
> > > | MCITrack-S224 | 73.8 | 85.6 | 40.7 |
> > > | ETBTrack(ODTrack-L384) [288,184] | **74.0** | **85.9** | **44.3** |
> > >
> > > It is evident that ETBTrack enables ODTrack to outperform different variants of MCITrack and SMTrack in both performance and efficiency under specific token budgets. Notably, ETBTrack requires only a single set of parameters, whereas other trackers necessitate distinct parameters for each variant. This highlights the flexibility of ETBTrack in supporting various computational environments, thereby cutting maintenance and development costs.
> > >
> > > ### To Q2
> > > We would like to clarify the principles of our design: Inference Efficiency and Discriminative Capability.
> > >
> > > Inference Efficiency. To minimize the computational footprint, our policy network reuses the tracker's intermediate encoder features, avoiding redundant computation. We strategically position the network early in the encoder, as it significantly reduces the computational cost of all subsequent Transformer layers.
> > >
> > > Discriminative Capability. The input to the policy network must contain sufficient global context to effectively distinguish between important and unimportant tokens. For instance, applying it to raw patch embeddings would fail to yield meaningful importance scores due to the lack of sufficient feature interaction. Consequently, we position the policy network at a depth where the encoder has generated sufficient global representations for token importance estimation.
> > >
> > > Guided by these principles, our design is backbone-agnostic. Extensive empirical results across various frameworks, resolutions, and backbone capacities confirm its generalizability. Furthermore, these principles underpin the performance-speed trade-off, which we have also analyzed in the ablation studies of Table 4.
> > >
> > > ### To Q3
> > > We will carefully revise the manuscript to improve readability. To assist in your assessment, we provide a concise summary of the design philosophy below:
> > >
> > > 1. **Motivation**.
> > > Our core objective is to achieve elastic token inference by identifying the most critical tokens for tracking. This allows the tracker to adapt to diverse computational environments using a single set of model parameters  by adjusting the token budget. The design focuses on two critical aspects: identifying tokens most important for localization and ensuring robust tracking performance under diverse token budgets.
> > >
> > > 2. **Token Importance Estimation via Reinforcement Learning**.
> > > Traditional token importance criteria often fail to align with the objective of tracking robustness, resulting in sub-optimal performance, as evidenced in Table 7 of the manuscript and the second table of our original rebuttal to Q1. To address this, we formulate token importance estimation as an reinforcement learning task. By evaluating the contribution of different token sets to precise target localization, we can directly align the importance criteria with the goal of robust tracking. Specifically, we sample different token sets to localize the target object. A high reward is assigned to token sets that achieve precise localization, while a low reward is assigned to those resulting in localization drift. Then, we employ a trainable policy network to capture these reward patterns and optimize it to predict higher importance scores to tokens with higher rewards, thereby ensuring that high-importance tokens are directly aligned with precise tracking outcomes.
> > >
> > > 3. **Robust Tracking via Collaborative Training**.
> > > Standard tracker parameters are generally not optimized for sparse token inference. To ensure robust performance across various token budgets and avoid bias, we employ a collaborative training strategy. Specifically, during training, the policy network selects the most important tokens under multiple budget constraints. The tracker is then optimized under these varying conditions simultaneously to ensure that the model generalizes across different token budgets rather than being limited to a specific one.
> > >
> > > 4. **Iterative Optimization**.
> > > These two optimization processes (policy network learning and tracker parameter optimization) are performed iteratively, allowing the components to mutually enhance each other’s performance and further improve the overall capability of the system.
> > >
> > > We are committed to incorporating these clarifications into the final version of the manuscript. We would be happy to provide further information if you have any additional questions.

---

### Official Review · Reviewer_wUKr · 2026-03-11

**Soundness:** 3
**Presentation:** 3
**Significance:** 2
**Originality:** 2
**Overall Recommendation:** 3
**Confidence:** 4

**Summary:**

This paper explores to enable visual trackers to operate under varying computational budgets without retraining separate models. The authors present a notable area of research by proposing ETBTrack, an elastic token budget training framework that allows a tracker to perform inference under different token budgets. The method introduces a policy network for token importance estimation and a collaborative optimization strategy to train trackers under multiple budgets. Experiments are conducted on Transformer-based trackers such as SeqTrack and ODTrack to demonstrate the effectiveness of the approach.

**Compliance With Llm Reviewing Policy:**

Affirmed.

**Key Questions For Authors:**

See weakness for the rebuttal. My main concern lies in the lack of SOTA tracker comparison.

**Limitations:**

yes

**Strengths And Weaknesses:**

Strength:
- The paper is well written and easy to follow;
- The paper targets an important problem in tracking systems: enabling a single model to adapt to different computational budgets across deployment scenarios;
- Experiments are conducted on Transformer-based trackers such as SeqTrack and ODTrack.

Weakness:
- The main concern is the lack of comparison with state-of-the-art trackers. The experimental comparison is mainly limited to token pruning variants of the same backbone tracker. However, the paper does not compare against strong SOTA trackers. Although the paper focuses on improving efficiency through elastic token budgets, it would still be important to compare with the other trackers. In practice, learning token importance and dynamically pruning tokens may remove redundant or noisy tokens, which can sometimes even improve tracking accuracy rather than merely reduce computation. Therefore, without comparisons against strong full-budget trackers, it is difficult to determine whether the proposed approach truly provides a favorable performance–efficiency trade-off.
- The paper claims that elastic token budgets enable better efficiency–accuracy trade-offs, but the experiments do not clearly demonstrate whether the resulting performance is competitive with existing efficient trackers or lightweight tracking architectures.
- The motivation of the paper emphasizes adapting trackers to diverse computational environments. Since elastic token budgets are particularly relevant for resource-constrained platforms, evaluating the method on mobile devices (or providing latency measurements on mobile GPUs/NPUs) would significantly strengthen the current work.

---

> ### Author Rebuttal · Authors · 2026-03-31
>
> **To Q1**
>
> We have conducted a performance comparison with representative trackers that utilize similar backbones or architectures.
>
> | Tracker | LaSOT (AUC) | TrackingNet (AUC) | FPS |
> | :--- | :---: | :---: | :---: |
> | HIPTrack | 72.7 | 84.5 | 42.6 |
> | ARTrack-384 | 72.6 | 85.1 | 13.2 |
> | TATrack-L | 71.1 | 85.0 | 7.2 |
> | AQATrack-384 | 72.7 | 84.8 | 43.8 |
> | ROMTrack-384 | 71.4 | 84.1 | 28.0 |
> | EVPTrack-384 | 72.7 | 84.4 | 29.6 |
> | **ODTrack-L384** | **74.0** | **86.1** | 34.8 |
> | **ETBTrack(ODTrack-L384) [288, 184]** | **74.0** | **85.9** | **44.3** |
> | **ETBTrack(ODTrack-L384) [288, 92]** | 72.1 | 84.7 | **52.1** |
> | **ETBTrack(ODTrack-L384) [144, 184]** | 72.6 | 85.2 | **51.5** |
> *(The notation $[B_z, B_x]$ denotes the elastic token budgets for templates and search regions, respectively.)*
>
> 1. **Efficiency Gain without Compromising Accuracy:** As demonstrated in the table, `ETBTrack_ODTrack [288, 184]` maintains the high accuracy of the original ODTrack (74.0% vs. 74.0% on LaSOT, 85.9% vs. 86.1% on TrackingNet) while providing a significant boost in inference speed (from 34.8 to 44.3 FPS). This confirms our claim that ETBTrack effectively eliminates redundant tokens with little tracking precision sacrifice.
> 2. **Superior Trade-off:** By comparing ETBTrack with other strong full-budget trackers, it is evident that ETBTrack offers a more flexible Pareto-optimal frontier. While other trackers are constrained to fixed computational budgets, our framework enables a single model to adapt to varying computational demands while consistently delivering competitive tracking results. By dynamically adjusting the token budget, ETBTrack achieves a superior performance-efficiency trade-off, enabling ODTrack to simultaneously outperform other methods in **both accuracy and computational efficiency**.
>
> **To Q2**
>
> We have conducted a systematic performance comparison between our ETBTrack and various state-of-the-art token pruning methods.
>
> We implemented existing pruning strategies (based on the SeqTrack-B256) following their original designs and compared them with our approach under different token configurations $[B_z, B_x]$. As shown in the table below, **ETBTrack consistently achieves superior performance across all budget settings**, demonstrating its effectiveness in maintaining high accuracy under elastic constraints.
>
> | SeqTrack-B256 | [32, 128] | [128, 128] | [128, 64] | **Average** |
> | :--- | :---: | :---: | :---: | :---: |
> | **ETBTrack (Ours)** | **65.8** | **69.2** | **66.1** | **67.03** |
> | VisionZip | 55.8 | 57.6 | 56.4 | 56.60 |
> | EViT | 54.7 | 57.4 | 56.7 | 56.27 |
> | ToMe | 54.9 | 57.1 | 55.3 | 55.77 |
> | OSTrack | 47.2 | 49.3 | 46.2 | 47.57 |
> | FastV | 31.5 | 35.3 | 32.9 | 33.23 |
> | LMTrack | 28.1 | 28.6 | 29.5 | 28.73 |
> | DivPrune | 9.3 | 12.4 | 10.2 | 10.63 |
> *Note: $[B_z, B_x]$ denotes the token budget for the template and search region, respectively. Bold indicates the best performance.*
>
>
>
> **To Q3**
>
> we have conducted extensive efficiency measurements on the NVIDIA Jetson Orin NX (16GB RAM), which is a representative mobile-computing platform. The hardware configuration includes a 1024-core NVIDIA Ampere Architecture GPU and an ARM Cortex-A78AE CPU.
>
> | Tracker | Token Budget | CPU FPS | GPU FPS |
> | :--- | :--- | :---: | :---: |
> | **SeqTrack_B256** | [512, 256] | 0.73 | 4.29 |
> | **ETBTrack_SeqTrack_B256** | [256, 128] | 1.30 | 7.41 |
> | | [128, 128] | 1.58 | 8.77 |
> | | [64, 128] | 1.75 | 9.80 |
> | | [32, 128] | 1.83 | 10.20 |
> | | [16, 128] | 1.85 | 10.42 |
> | | [256, 64] | 1.47 | 8.33 |
> | | [128, 64] | 1.78 | 9.71 |
> | | [64, 64] | 1.86 | 10.53 |
> | | [32, 64] | 2.11 | 10.99 |
> | | [16, 64] | 2.23 | 11.24 |
>
> As shown in the above table, ETBTrack allows for significant adjustments to tracking efficiency (up to 2.9$\times$ FPS) on mobile devices by varying the token budget. These results validate the capability of our proposed ETBTrack to meet the diverse computational requirements of mobile devices. Due to space constraints, we only present the efficiency analysis for SeqTrack-B256 here. The efficiency measurements for other base model will be included in the final version of the manuscript.
>
> All of the above discussions will be included in the manuscript.

---

> > ### Author Rebuttal · Reviewer_wUKr · 2026-04-06
> >
> > The authors partially addressed my concerns. However, my main concern still lies in the SOTA comparison. The paper currently lacks comparisons with more recent and complete trackers. For example, while ARTrack-384 is included in Q1, it is unclear why the more recent ARTrackV2 is not considered, given that it demonstrates both improved performance and efficiency over ARTrack. Overall, the paper appears to focus primarily on self-improvement over its own baselines. While such ablations are valuable, it is essential for a paper targeting top-tier venues to include comprehensive and up-to-date SOTA comparisons. Without sufficiently benchmarking against recent strong methods, it is difficult to accurately assess the true contribution and competitiveness of the proposed approach.

---

> > > ### Author Response · Authors · 2026-04-06
> > >
> > > We thank the reviewer for the constructive suggestion regarding the state-of-the-art comparisons.
> > > To address your concerns, we have expanded our comparison with a comprehensive table, which includes more strong and efficient trackers such as **ARTrackV2**[1], LoRAT[2], SMTrack[3] and MCITrack[4].
> > >
> > > | Method | LaSOT (AUC) | TrackingNet (AUC) | FPS |
> > > | :--- | :---: | :---: | :---: |
> > > | HIPTrack | 72.7 | 84.5 | 42.6 |
> > > | ARTrack-384 | 72.6 | 85.1 | 13.2 |
> > > | TATrack-L | 71.1 | 85.0 | 7.2 |
> > > | AQATrack-384 | 72.7 | 84.8 | 43.8 |
> > > | ROMTrack-384 | 71.4 | 84.1 | 28.0 |
> > > | EVPTrack-384 | 72.7 | 84.4 | 29.6 |
> > > | ARTrackV2-L384 | 73.6 | 86.1 | 43.2 |
> > > | LoRAT-g-224 | 74.9 | 85.2 | 41.6 |
> > > | SMTrack-M384 | 71.9 | 85.2 | 34.0 |
> > > | MCITrack-S224 | 73.8 | 85.6 | 40.7 |
> > > | ODTrack-L384 | 74.0 | 86.1 | 34.8 |
> > > | ETBTrack(ODTrack-L384) [288, 184] | 74.0 | 85.9 | 44.3 |
> > > | ETBTrack(ODTrack-L384) [288, 92] | 72.1 | 84.7 | 52.1 |
> > > | ETBTrack(ODTrack-L384) [144, 184] | 72.6 | 85.2 | 51.5 |
> > > *(The notation $[B_z, B_x]$ denotes the elastic token budgets for templates and search regions, respectively.)*
> > >
> > > We would like to clarify our contributions through the following points:
> > >
> > > + **Beyond Self-Improvement**: Beyond our initial validation with baselines such as ODTrack, we have now conducted extensive benchmarking against a wide range of recent state-of-the-art trackers to provide a more comprehensive evaluation. The results indicate that ETBTrack enables ODTrack to dynamically achieve performance and efficiency levels that match or exceed specialized models. For instance, `ETBTrack(ODTrack-L384) [288, 184]` achieves an AUC comparable to ARTrackV2 while offering higher inference speed.
> > >
> > > + **Evolution of Tracking Paradigms**: Methods like ARTrackV2 achieve their performance through end-to-end architectural redesign. In contrast, our contribution lies in the post-training optimization framework. ETBTrack not only pushes the performance-efficiency Pareto front of existing baselines to the state-of-the-art level, but also enables robust elastic computation, representing a fundamental improvement to the current tracking paradigm.
> > >
> > > + **Strengthening the Experimental Evaluation**: By including these recent trackers, we provide a more rigorous assessment to clarify the true contribution and competitiveness of our proposed approach. This broader context demonstrates that ETBTrack is a highly competitive and versatile solution for real-world scenarios where computational demands fluctuate.
> > >
> > > Overall, a fundamental question addressed by this study is how to enable existing trackers to robustly adapt to diverse real-world computational constraints using a unified set of parameters. The comprehensive experimental comparisons demonstrate that ETBTrack goes beyond mere self-improvement on its own baselines; it effectively pushes the performance-efficiency Pareto front of these baselines toward the state-of-the-art level. The paradigm shift introduced by ETBTrack offers broader applicability, making it a highly versatile solution for real-world scenarios with fluctuating computational demands.
> > >
> > > We sincerely appreciate your valuable comments, as they have significantly helped us improve the rigor of our work. We are committed to incorporating all the aforementioned comprehensive comparisons and detailed discussions into the final version of our manuscript. We remain at your disposal for any further questions or discussions.
> > >
> > > [1] ARTrackV2: Prompting Autoregressive Tracker Where to Look and How to Describe
> > >
> > > [2] Tracking Meets LoRA: Faster Training, Larger Model, Stronger Performance
> > >
> > > [3] SMTrack: State-Aware Mamba for Efficient Temporal Modeling in Visual Tracking
> > >
> > > [4] Exploring Enhanced Contextual Information for Video-Level Object Tracking

---

### Official Review · Reviewer_FviF · 2026-03-12

**Soundness:** 3
**Presentation:** 3
**Significance:** 4
**Originality:** 3
**Overall Recommendation:** 4
**Confidence:** 3

**Summary:**

This paper provides the first exploration of the elastic token budget training framework, enabling trackers to perform robust tracking under varying computational budgets.
ETBTrack employs result-driven importance criteria, sampling multiple groups of template tokens and drawing inspiration from GRPO to compute advantages and optimize the policy networks that score token importance.
ETBTrack collaboratively optimizes the tracker across different budgets by utilizing the most critical tokens identified by the policy network, thereby enabling the tracker to operate effectively under diverse token budgets.
The paper conducts experiments on LaSOT, LaSOT-ext, and TrackingNet, using SeqTrack and ODTrack as backbones. The results demonstrate that when the token budget is reduced, ETBTrack can effectively preserve the original performance of the tracker.

**Compliance With Llm Reviewing Policy:**

Affirmed.

**Final Justification:**

The rebuttal largely addressed my concerns. I’ll keep the score at 4.

**Key Questions For Authors:**

Please refer to the weaknesses listed above.

**Limitations:**

Yes.

**Strengths And Weaknesses:**

### Strengths：

- This paper provides the first exploration of the elastic token budget training framework, enabling trackers to perform robust tracking under varying computational budgets.

- The paper adopts a result-driven importance criterion to train the policy network, which is more reasonable than approaches based on attention scores or similar heuristics.

- Extensive experiments are conducted on multiple benchmarks using two different backbone models, which convincingly demonstrate the effectiveness of the proposed method.

### Weakness：

- Does the policy network learn interpretable patterns of token importance? For example, does it tend to preserve tokens corresponding to object boundaries or texture-rich regions?

---

> ### Author Rebuttal · Authors · 2026-03-30
>
> We appreciate the reviewer's insightful question regarding the interpretability of the policy network. As demonstrated in Figure 3(b) of the manuscript and Figure 1 in the Supplementary Materials, the policy network indeed learns highly interpretable patterns of token importance.
>
> Specifically, the network consistently assigns higher importance scores to tokens corresponding to the target objects and texture-rich regions. These tokens provide the most critical semantic and texture cues for precise regression of the target bounding box. By preserving these salient tokens while pruning redundant background information, the policy network effectively balances computational efficiency with localization precision.

---

> > ### Author Rebuttal · Reviewer_FviF · 2026-04-02
> >
> > Thank you for your response. I will keep my score as weak accept.

---

### Decision · Program_Chairs · 2026-04-30

**Decision:**

Accept (regular)

**Comment:**

The paper initially received mixed reviews 3343. The major concerns were:

1) Are interpetable patterns of token importance learned? [FviF]
2) lacks comparison to full-budget SOTA trackers (only compares variants of the same backbone) [wUKr]
3) lacks comparisons with existing efficient or lightweight trackers [wUKr]
4) evaluation on mobile devices is missing (to support claim of diverse computational environments) [wUKr]
5) Presentation issues [w21s]
6) generalizability of the policy layer? [w21s]
7) how does it compare to reducing the resolution and distilling the model? [w21s]
8) retraining is required, which increases cost [nnP4]
9) lacks comparison to standard methods [nnP4]
10) missing comparisons to training-free token reduction methods [nnP4]

The authors wrote a response to address the concerns. After the discussion period:
- FviF (rating 4) was satisfied.
- wUKr (rating 3) was partially satisfied, but still thought additional comparisons were needed (e.g., with ARTrackV2). The authors provided further additional results, showing that the proposed method achieved similar AUC and FPS as ARTrackV2 and LoRAT.
- w21s (rating 3) was partially satisfied, and still concerned about the missing performance with low-latency trackers, and the lack of principle discussion about the policy layer's insertion.  The authors provided further experiment results demonstrating the superiority to MCITrack and SMTrack. In addition, the authors noted that their method only uses a single set of parameters and can adapt to different computation budgets easily.
- nnP4 was satisfied, and increased their score from 3 to 4.

After the discussion period, the final ratings were mixed 3344. After reading the paper and discussions, the AC thought that the concerns were all addressed well. While wUKr and w21s were still concerned about the comparison with low-latency SOTA trackers, this comparison was provided later and looks promising. While the method is similar to ARTrackV2 in performance, the AC notes that the proposed work has two important abilities: 1) it can change the number of tokens using the same set of parameters is practically useful for environments with dynamic computional demands; 2) the proposed method is generally applicable to any transformer-based tracker. Regarding w21s's concern about the policy network placement, the authors provided more reasoning.

Overall, the paper makes a solid contribution to elastic budget token-pruning, as a general framework, and could see interest in the tracking community. Thus the AC recommends accept. The authors should revise the paper according to the reviews, rebuttal, and discussion.